# The Effect of Yeast Fermentation of Two Lupine Species on the Digestibility of Protein and Amino Acids, Microflora Composition and Metabolites Production in the Ileum of Growing Pigs

**DOI:** 10.3390/ani11102894

**Published:** 2021-10-04

**Authors:** Małgorzata Kasprowicz-Potocka, Anita Zaworska-Zakrzewska, Marcin Taciak, Andrzej Frankiewicz

**Affiliations:** 1Department of Animal Nutrition, Faculty of Veterinary Medicine and Animal Sciences, Poznan University of Life Sciences, Wołyńska 33, 60-637 Poznan, Poland; anita.zaworska-zakrzewska@up.poznan.pl (A.Z.-Z.); andrzej.frankiewicz@up.poznan.pl (A.F.); 2Department of Animal Nutrition, The Kielanowski Institute of Animal Physiology and Nutrition, Polish Academy of Sciences, Instytucka 3, 05-110 Jabłonna, Poland; m.taciak@ifzz.pl

**Keywords:** *Candida utilis*, fermentation, lupine, pigs, digestibility, microflora

## Abstract

**Simple Summary:**

The fermented feed component in pigs‘ diet may influence the microbiota of the gastrointestinal tract, improve the use of nutrients from the diet, and reduce the level of excreted N and P. The aim of this study was to investigate the effects of raw and *Candida utilis*-fermented yellow and narrow-leaved lupine seeds on the apparent ileal digestibility coefficients of protein and amino acids and the metabolic activity of the intestinal microflora in five cannulated male pigs. Fermentation significantly improved the ileal digestibility of protein, asparagine, threonine, tyrosine, histidine, and arginine in the lupine seeds and increased the counts of total bacteria and yeast, the pH value, isobutyrate and isovalerate concentrations in the ileal digesta, but decreased the dry matter and ammonia content. The narrow-leaved lupine seeds were characterized by higher digestibility of asparagine, threonine, serine, alanine, valine, isoleucine, and arginine. The digesta of the pigs fed with these seeds had higher counts of lactic acid bacteria and moulds but lower total bacteria count than the digesta of the pigs fed with yellow lupine seeds.

**Abstract:**

The aim of this study was to investigate the effects of raw and *Candida utilis*-fermented yellow (YL) and narrow-leaved lupine (NL) seeds on the apparent ileal digestibility coefficients (AID) of protein and amino acids in pigs and the metabolic activity of their intestinal microflora. Five cross-bred castrated 25-kg barrows were surgically fitted with a T-cannula in the distal ileum and housed individually in metabolic cages. They were fed five semi-synthetic diets containing only one source of protein: soybean meal (SBM), raw or fermented yellow lupine seeds (RYL or FYL), raw or fermented narrow-leaved lupine seeds (RNL or FNL). The study period consisted of six-day adaptation to the diet and one-day collection of digesta, which was sampled for microbial and chemical analyses. The AID coefficients of protein and amino acids were calculated with the marker method with TiO_2_. One-way (feed effect) and two-way (variety effect, fermentation effect) analysis of variance (ANOVA) with Duncan’s test at *p* < 0.05 were applied. The digesta from the SBM and FNL variants had significantly higher (*p* < 0.05) AID coefficients of protein, asparagine, threonine, serine, isoleucine, leucine, histidine, and tyrosine, whereas the SBM variant was characterized by the lowest AID of cystine and the highest AID of alanine and methionine (*p* < 0.05). The ileal digesta of the pigs fed with FYL contained more bacteria, whereas the count of yeast was higher in the FNL variant. The digesta of the pigs from the FNL and RYL variants had the highest count of moulds (*p* < 0.05), whereas the digesta in the FYL, SBM, and RNL variants had almost no moulds at all. The ileal dry matter content was significantly lower in SBM group. The lowest pH was noted in the RYL variant (*p* < 0.05). The content of ammonia and total volatile fatty acids in the ileal digesta of the SBM variant was the lowest (*p* < 0.05). Fermentation significantly improved the AID of protein, asparagine, threonine, tyrosine, histidine, and arginine, increased the counts of total bacteria and yeast, the pH value, and isobutyrate and isovalerate concentrations, but decreased the dry matter and ammonia content (*p* < 0.05). The digesta of the NL variants was characterized by higher AID of asparagine, threonine, serine, alanine, valine, isoleucine, and arginine, and higher counts of LAB and moulds but lower total bacteria count than in the YL variants. The ileal pH was lower in the YL variants, where higher isobutyrate and butyrate concentrations (*p* < 0.05) were observed. To sum up, fermentation increased the counts of the ileal microbiota and improved the digestibility coefficients of protein and some amino acids. The narrow-leaved lupine seeds resulted in more positive changes in the digesta of growing pigs than yellow lupine.

## 1. Introduction

Fermented feeds have become popular as good quality components of diets and as functional food which could be beneficial for consumer health [1,2,3]. Fermented products contain not only proteins, minerals, and vitamins but also living or dried cells of lactic acid bacteria or yeast, which can affect probiotically or prebiotically the digestive tract microflora [4,5]. The supplementation of the diet with fermented components has beneficial effect on the growth of pigs and regulation of the intestinal microflora because it improves the digestibility of protein and amino acids, and it reduces the concentration of antinutritional factors [3,6,7,8,9]. Currently, soybean meal (SBM) is a basic high-protein component used in the production of complete diets for pigs. However, as the climate in Europe is rather unsuitable to grow soy; soybean meal is usually imported, so its price is high. The EU is concerned about the possibilities of reducing the SBM level in diets for pigs and poultry [10,11,12,13]. Legume seeds and their products could replace soybean meal totally or partially, but unprocessed seeds contain many antinutritional substances (ANFs) such as alkaloids, phytate, and oligosaccharides, which may negatively influence the availability and utilization of nutrients by monogastric animals [14,15,16]. Fermentation with different microorganisms can reduce ANFs to a safe level. The products are characterized by acidic pH and the dominance of lactic and propionic acids, which are beneficial for the gastrointestinal microflora, especially in young pigs [14,17]. After fermentation, lupine seeds are more suitable for animals and they could replace SBM in the diet. So far there have been only a few studies on the nutritional value and physiological properties of fermented lupine seeds used for animal nutrition [14,17,18]. The following research hypothesis was assumed: the fermentation of lupine seeds improves their nutritional value, increases the digestibility of nutritional components and positively influences the ecosystem of the pig’s gastrointestinal tract. The ileal digestibility coefficients of amino acids are also necessary to calculate the right balance of amino acids in the feed ration for pigs. This study presents the effects of raw and *Candida utilis*-fermented yellow and narrow-leaved lupine seeds on the pigs’ digestibility coefficients of amino acids and protein in the ileum and the metabolic activity of the pigs’ intestinal microflora.

## 2. Materials and Methods

### 2.1. Lupine Seeds and Fermentation Process

The seeds of *Lupinus luteus* (var. Lord) and *Lupinus angustifolius* (var. Graf) were obtained from the Plant Breeding Station in Przebędowo, Poland. The *Candida utilis* strain was bought from the Pure Culture Collection of Industrial Microorganisms LOCK 105 at the Institute of Fermentation Technology and Microbiology, Łódź University of Technology, Poland. Fermented yellow and narrow-leaved lupine seed products were obtained in the process described by Kasprowicz-Potocka et al. [7]. The raw seeds and fermented products were characterized in Table 1 and Table 2. 

### 2.2. Experiment on Animals

All the experimental procedures complied with the guidelines of the Local Ethical Committee for Experiments on Animals in Poznań (43/2011) regarding animal experimentation and care and they followed the EU Directive 2010/63/EU on the protection of animals used for scientific purposes. The pigs received all the necessary veterinary vaccinations and had unlimited access to water.

Five cross-bred castrated barrows with an average initial body weight of about 25 kg were surgically fitted with a T-cannula in the distal ileum, following the method described by van Leeuwen et al. [19]. After ten-day recuperation each of the pigs was housed in an individual metabolic cage and was randomly allotted to a split-plot model with five diets and five experimental periods. All diets were semi-synthetic and contained only one source of protein: soybean meal (SBM), raw yellow (RYL) or fermented yellow lupine seeds (FYL), raw narrow-leaved (RNL) or fermented narrow-leaved lupine seeds (FNL) at an amount resulting from the protein content in the components (Table 3). The diets were supplemented with starch, sugar, and soybean oil. All diets also contained 3 g/kg titanium dioxide as an indigestible marker. The diets had similar energy and protein content. Vitamins and minerals were included in all diets to meet the current requirement estimates of GfE [20]. Dry feed was given twice a day (8.00; 16.00). The study period consisted of six-day adaptation to the diet and one-day collection of digesta. The digesta collected from each pig was mixed separately and representative samples were packed into the plastic bags. After the last sampling, fresh digesta was analysed microbiologically and the ileal pH was measured in each sample. The other fresh samples were frozen for chemical analysis (dry matter, ammonia, volatile fatty acids (VFA)). The rest of the collected material was frozen at −70 °C and lyophilized. The dry matter and protein content, the amino acid composition of protein, and the titanium dioxide content in the digesta samples were determined.

The apparent ileal digestibility (AID) coefficients of protein and amino acids in SBM, raw and fermented lupine seeds were calculated with the following formula:AID (%) = 100 − 100 × (% of marker in feed/% of marker in faeces) × (% nutrient in faeces/% of nutrient in feed).

### 2.3. Chemical and Microbial Analyses

For chemical analysis all the samples were ground to pass through a 0.5-mm sieve. Raw seeds, fermented products, and digesta samples (*n* = 5, from different animals) were analysed in duplicate. The content of dry matter (DM), crude protein (CP), crude ash (CA), acid detergent fibre, and neutral detergent fibre was measured with methods 934.01, 976.05, 920.39, 942.05, and 973.18, respectively, according to the AOAC [21]. Lupine alkaloids were extracted from flour with trichloroacetic acid and methylene chloride (Sigma-Aldrich, Munich, Germany). The content of alkaloids was measured by means of gas chromatography (Shimadzu GC17A, Kyoto, Japan) with a capillary column (Phenomenex, Torrance, CA, USA). Raffinose family oligosaccharides were extracted and analysed by means of high-resolution gas chromatography, as described by Zalewski et al. [22]. The phytate content was measured according to the method described by Haug and Lantzsch [23]. The amino acid (AA) content was measured with an AAA-339 Mikrotechna amino acid analyser (Prague, Czech Republic), using ninhydrin for post-column derivatization. Before analysis, the samples were hydrolysed with 6 M HCl for 24 h at 110 °C, according to procedure 994.12; AOAC [21]. The level of titanium dioxide (POCH, Gliwice, Poland) was measured with the method described by Short et al. [24]; the samples were prepared in accordance with the procedure described by Myers et al. [25]. Ten grams of the mixed ileal sample was added to 90 mL of ion-exchanged water and shaken.

The pH was measured in the liquid phase with a pH meter (model 301, Hanna Instruments, Vila do Conde, Portugal). Ammonia was analysed with the spectrometric method and Nessler reagent (POCh, Gliwice, Poland). The VFAs were analysed according to the procedure described by Barszcz et al. [26] with an HP 5890 Series II gas chromatograph (Hewlett Packard, Waldbronn, Germany) with a flame-ionization detector and a SupelcoNukol (Supelco, Bellafonte, KY, USA) fused silica capillary column (30 m × 0.25 mm i.d.; 0.25 mm). Helium was the carrier gas.

Samples for bacteriological analysis were prepared by adding 27 mL of buffered peptone water (Oxoid) to 3 g of samples and homogenizing for 30 s in a laboratory stomacher. Microbial counts were measured with a decimal dilution series of the homogenized samples. The total bacteria count and lactic acid bacteria count were measured with plate standard methods, using plate count agar (PCA) and MRS broth (Oxoid), respectively, after 72-h incubation at 30 °C. The yeast content was calculated with pre-supplemented DRBC (Oxoid) after incubation at 25 °C for 3 to 5 days. The count of coliform bacteria was measured with Violet Red Bile lactose agar (Oxid, Hampshire, UK) after 24-h incubation at 30 °C.

### 2.4. Statistical Analysis

The SAS Enterprise Guide 5.1 (USA) program was used for statistical analysis. The data were analysed with one-way (feed effect) and two-way (variety effect, fermentation effect) analysis of variance ANOVA. The differences between the means were compared with Duncan’s test at *p* < 0.05.

## 3. Results

The AID coefficients of protein and amino acids (Table 4) were different in almost all components under analysis, except for glutamine, proline, glycine, phenylalanine, and lysine. The SBM and FNL were characterized by significantly higher (*p* < 0.05) AID coefficients of crude protein, asparagine, threonine, serine, isoleucine, leucine, histidine ant tyrosine than the other variants. The digesta of the pig from the SBM variant had lower AID of cysteine than in both lupine groups, but the AID coefficients of alanine and methionine were significantly higher (*p* < 0.05) than in the other components. Fermentation significantly improved the digestibility coefficients of crude protein, asparagine, threonine, tyrosine, histidine, and arginine, whereas narrow-leaved lupine was characterized by higher apparent ileal digestibility coefficients of asparagine, threonine, serine, alanine, valine, isoleucine, and arginine than yellow lupine. There was interaction between the seeds and fermentation process in the AID of threonine, serine, leucine, and histidine.

The ileal digesta of the pigs fed with FYL (Table 5) contained more bacteria (*p* < 0.05) than in the other variants. The yeast count was significantly higher (*p* < 0.05) in the FNL than in the SBM and RYL digesta. The yeast count in the FYL variant was also significantly higher than in the RYL digesta. The highest count of moulds was found in the digesta of the pigs from the FNL and RYL variants (*p* < 0.05), whereas the FYL, SBM and RNL digesta had almost no moulds. There were no differences in the LAB and *Enterobacteriaceae* content (*p* > 0.05). Fermentation significantly increased the counts of total bacteria and yeast, whereas the ileal digesta of the pigs fed with narrow-leaved lupine products had higher LAB and moulds counts but lower total bacteria count. There was interaction between the counts of total bacteria, moulds, and yeast.

The ileal digesta of the pigs from all variants contained different (*p* < 0.05) amounts of dry matter, ammonia, and volatile fatty acids (except propionate and valerate) (Table 6). The dry matter content in the ileal digesta in the RNL, FNL, RYL and FYL variants was significantly higher than in the SBM. The RNL digesta was characterized by higher dry matter content than the FNL. The lowest pH was noted in the ileal digesta of the pig fed with RYL (*p* < 0.05). There was significantly higher pH in the digesta of both NL variants. The ammonia content in the ileal digesta of the SBM variant was the lowest (*p* < 0.05), whereas in the RYL and RNL variants it was significantly higher (*p* < 0.05) than in other variants. The lowest VFA content in the ileal digesta was found in the SBM variant; the highest—in the FNL and RYL variants. The acetate content in the digesta of the pig fed with SBM was significantly higher (*p* < 0.05) than in the RNL and FYL variants. The isobutyrate and butyrate content in the ileal digesta of the FYL variant were higher than in the SBM, RNL, and FNL variants, whereas the isovalerate content in the FNL and FYL digesta was higher than in the other variants. Fermentation significantly (*p* < 0.05) decreased the dry matter and ammonia content in the digesta. Moreover, the digesta of the pigs fed with fermented seeds had higher pH (*p* < 0.05) and isobutyrate and isovalerate contents. The ileal digesta of the pigs fed yellow lupine seeds had lower pH but higher isobutyrate and butyrate contents (*p* < 0.05). There was no interaction between the seeds and fermentation.

## 4. Discussion

The yellow and narrow-leaved lupine seeds were fermented by *Candida utilis*. This strain was selected in this study as it proved to be the most efficient in earlier experiments on lupine seed fermentation with different yeast strains [15,27]. The fermentation of the seeds of both lupine species affected their chemical composition. It increased the protein, lysine, cystine, and threonine content but not the methionine content. Fermentation by *Candida utilis* also reduced the ADF, NDF, and phytate-P, but the content of alkaloids increased. This is consistent with the observations made by Kasprowicz-Potocka et al. [27], who concluded that lupine alkaloids were resistant to yeast fermentation. After the fermentation no RFO was found in the seeds, which was in line with the observations made by other authors [6,17,27,28,29]. It is clear that an increase in the crude protein content results from a decrease in the content of nonstructural and structural carbohydrates in the biomass and from the growth of microorganisms. Fermented products were characterized by acidic pH and higher content of yeast and bacteria. Other authors also observed a higher level of lactic acid bacteria, yeast, and total bacteria after the fermentation of legumes [17,30]. 

The experimental diets contained less fermented than raw seeds, which was reflected by the higher protein content in the fermented seeds, but the nutritional value of all the diets was similar. The digestibility of nutrients is an important factor determining the use of feed and it is indirectly related to the protein quality and the ANF level [31]. According to the data in scientific publications, the digestibility of SBM is slightly higher than that of lupine meal, which mostly results from the presence of anti-nutrients in lupine seeds [32]. In the current study, the AID of protein and amino acids in SBM was similar to the values reported by Kong et al. [33] and by Upadhaya and Kim [34]. The AID coefficient of fermented narrow-leaved lupine protein did not differ from that of SBM but in raw narrow-leaved as well as raw and fermented yellow lupine seeds the digestibility of protein was significantly lower. In a study on rats the apparent protein digestibility of fermented lupine seeds was about 2% greater than that of unprocessed seeds [27]. Zaworska-Zakrzewska et al. [17] conducted a study on pigs and found that the bacterial fermentation of narrow-leaved lupine seeds (var. Neptun) increased the digestibility of crude protein by 11%. In this study, it was only 1.2% and 7% for yellow lupine and narrow-leaved lupine, respectively. In the current research, fermentation significantly improved the digestibility of asparagine, threonine, tyrosine, histidine, and arginine, whereas Zaworska et al. [6] noted that fermented narrow-leaved lupine seeds had significantly higher AID coefficients for arginine, isoleucine, and cysteine than row seeds. The differences may have been caused by the higher content of protein in the fermented seeds and by the higher level of antinutrients such as alkaloids, RFOs, and phytate in the raw lupine seeds, especially yellow ones. Antinutrients inhibit the activity of protolithic enzymes and thus reduce the digestibility of feed [31,35]. Upadhaya and Kim [34] fermented SBM with bacteria and yeasts and observed significant improvement of pigs’ intestinal digestibility. Kim et al. [36] found that SBM fermented with *Bacillus subtilis* significantly improved the AID of phenylalanine, isoleucine, and valine. In the present study, narrow-leaved lupine seeds were characterized by higher digestibility coefficients of protein and amino acids than yellow lupine seeds. A similar trend was observed in the study on rats [27]. The differences in the AID coefficients for the protein and amino acids in SBM and in lupine seeds may have been caused by the conditions and methods of conducting research on animals (different breeds, indices, methodologies). They may also depend on the chemical characteristics of raw materials (cultivar, the level of nutrients, ANF content) and conditions of fermentation (microorganisms, time, humidity, temperature). 

The composition of a diet may affect not only the digestibility of nutrients but it may also influence the microflora. In the current research the fermented products contained more lactic acid bacteria and yeast than the raw seeds due to the microbial activity during processes. At the same time the pH in the fermented products was lower. The ileal digesta of the pigs consuming fermented products also had higher levels of total bacteria and yeast, but fermentation did not affect the content of moulds and enterobacteria. By contrast, the studies by Zaworska et al. [6] and Zaworska-Zakrzewska et al. [17] showed that fermented narrow-leaved lupine seeds did not affect the ecology of pigs’ gastrointestinal tract. There were also differences between cultivars, with higher counts of moulds and LAB in the digesta of the pigs fed with narrow-leaved lupine seeds in the diet. This effect may have been caused by the higher content of these microorganisms in the feed components as well as the higher content of narrow-leaved products in the diet. The characteristic pH value of about 6–7 in the pig’s ileum does not favour the development of lactic acid bacteria, but some other bacteria and yeast can grow, as was evidenced by this study. Yeast development is the most effective at pH of 4.5–6.5, but the activity connected with prebiotic functions generally takes place in the distal ileal part of the digestive tract. Moreover, other authors found that the supplementation of feed with fermented components improved the regulation of pigs and rats’ intestinal microflora [37,38]. Fermented components in the diet increased pH in the ileum, but decreased the dry matter level and ammonia content. It pointed to rather poor microbial activity in the ileal part of the digestive tract, which is in line with the low content of total volatile fatty acids. The protein structure in the fermented products was partly degraded by bacteria. The degradation may have made them more digestible, as was proved by the higher AID of protein and some amino acids. On the other hand, the lower content of dry matter, ammonia, and fatty acids in the soybean meal may have been caused by the lower count of microorganisms in the digesta. It is likely that the ileal digesta of the pigs fed with lupine seed meals had higher concentrations of short-chain fatty acids than the digesta of the pig fed with soybean meal due to the higher total content of lupine NSP and lupine oligosaccharides in the raw seeds. In the current study there were differences between the varieties. The yellow lupine products reduced the pH of the digesta by 0.4 and increased the isobutyrate and butyrate levels. The main end products formed during the fermentation of nondigestible carbohydrates are acetic, propionic, and butyric acids [39], as evidenced by the results of this study. Only the isovalerate and isobutyrate content was significantly higher in the ileal digesta of the pigs consuming fermented feed, but generally the amount of these two acids was very low. In the study by Zaworska et al. [6] the digesta of the pigs fed with raw narrow-leaved lupine seeds had significantly higher concentrations of total SCFA than the raw seeds. 

## 5. Conclusions

To sum up, fermentation reduced the content of antinutritional substances in the lupine seeds, which increased the content of protein and some amino acids. Moreover, it reduced the pH of the product and increased the counts of yeast and lactic acid bacteria. The chemical and biological changes in the seeds caused by fermentation increased the counts of bacteria and yeast in the digesta and thus improved the digestibility coefficients of protein and some amino acids as threonine, arginine, histidine and tyrosine in the ileum of pigs. Generally, the fermentation of narrow-leaved and yellow lupine seeds led to similar results, but fermented narrow-leaved seeds seemed to be more convenient for the pigs. Using of *Candida utilis* yeast for fermentation of narrow-leafed lupine seeds could be a good solution for improving of feed quality and its nutritional value, but it could increase the cost of the nutrition, so more research should be provided including economic analysis of feeding. 

## Figures and Tables

**Table 1 animals-11-02894-t001:** The chemical composition of raw and fermented lupine seeds.

Item %	Raw Seeds		Fermented Seeds	
	Narrow-Leaved Lupine	Yellow Lupine	Narrow-Leaved Lupine	Yellow Lupine
Dry matter	87.94 ± 0.02	88.6 ± 0.22	96.33 ± 0.04	94.99 ± 0.13
Crude protein	31.5 ± 0.6	37.3 ± 0.5	35.7 ± 0.2	47.0 ± 0.3
Lysine	1.29 ± 0.02	1.65 ± 0.01	1.54 ± 0.02	2.02 ± 0.01
Methionine	0.16 ± 0.01	0.20 ± 0.02	0.10 ± 0.01	0.17 ± 0.01
Cystine	0.36 ± 0.04	0.71 ± 0.02	0.64 ± 0.02	0.90 ± 0.01
Threonine	0.94 ± 0.05	1.08 ± 0.03	1.02 ± 0.01	1.24 ± 0.03
Crude ash	3.4 ± 0.1	4.2 ± 0.1	3.5 ± 0.1	4.5 ± 0.1
ADF	19.9 ± 0.2	18.7 ± 0.9	19.7 ± 0.5	16.1 ± 0.4
NDF	24.7 ± 0.1	24.5 ± 1.0	21.1 ± 0.7	18.9 ± 1.3
Total alkaloids	0.007 ± 0.001	0.016 ± 0.001	0.021 ± 0.003	0.042 ± 0.003
Total RFOs	6.6 ± 0.2	7.9 ± 0.3	0.0 ± 0.0	0.0 ± 0.0
Phytate phosphorus	0.36 ± 0.07	0.59 ± 0.03	0.12 ± 0.01	0.42 ± 0.09

The results are expressed as mean values ± standard deviation; ADF—acid detergent fiber; NDF—neutral detergent fiber; RFOs—oligosaccharides raffinose family.

**Table 2 animals-11-02894-t002:** The microbial status of raw and fermented lupine seeds.

Item CFU/g	Raw Seeds		Fermented Seeds	
	Narrow-Leaved Lupine	Yellow Lupine	Narrow-Leaved Lupine	Yellow Lupine
pH	5.5 ± 0.1	5.6 ± 0.2	4.11 ± 0.1	3.98 ± 0.1
Total bacteria	5.4 × 10^5^ ± 1.1 × 10^2^	4.6 × 10^5^ ± 1.4 × 10^3^	2.5 × 10^6^ ± 1.4 × 10^3^	8.4 × 10^5^ ± 1.1 × 10^3^
Lactic acid bacteria	2.4 × 10^4^ ± 7.1 × 10^2^	3.9 × 10^4^ ± 1.2 × 10^2^	4.8 × 10^5^ ± 3.1 × 10^3^	7.0 × 10^5^ ± 1.1 × 10^3^
Yeast	2.3 × 10^4^ ± 9.9 × 10^2^	3.7 × 10^3^ ± 1.1 × 10^2^	4.0 × 10^6^ ± 1.1 × 10^4^	3.4 × 10^6^ ± 2.1 × 10^3^
Coliform bacteria	5.5 × 10^4^ ± 2.4 × 10^2^	7.5 × 10^4^ ± 4.2 × 10^2^	1.5 × 10^3^ ± 1.4 × 10^2^	1.5 × 10^4^ ± 9.1 × 10^2^

**Table 3 animals-11-02894-t003:** The composition and calculated nutritional value of diets.

Components %	SBM	RNL	FNL	RYL	FYL
Soybean meal	35.00	-	-	-	-
Raw yellow/narrow-leaved lupine seeds	-	51.38	-	42.00	-
Fermented yellow/narrow-leaved lupine seeds	-	-	42.00	-	39.00
Maize starch	51.51	32.00	43.18	41.38	46.18
Sugar	10.00	10.00	10.00	10.00	10.00
Soya oil	0.00	3.00	1.00	3.00	1.00
Limestone	0.70	0.70	0.70	0.70	0.70
Calcium phosphate	1.70	1.80	2.00	1.80	2.00
NaCl	0.29	0.32	0.32	0.32	0.32
Premix *	0.50	0.50	0.50	0.50	0.50
Titanium dioxide	0.30	0.30	0.30	0.30	0.30
ME MJ/kg	14.3	14.1	14.1	14.1	14.2
Crude protein %	16.5	16.3	16.2	16.2	16.4
Ca %	0.86	0.86	0.88	0.88	0.87
P %	0.62	0.60	0.62	0.62	0.61
Na %	0.13	0.13	0.13	0.13	0.13

SBM—diet contained soybean meal as the sole component high in protein; RYL—diet contained raw yellow lupine seeds; FYL—diet contained fermented yellow lupine seeds; RNL—diet contained raw narrow-leaved lupine seeds; FNL—diet contained fermented narrow-leaved lupine seeds; ME—metabolizable energy, was calculated in the program WinPasze, based on chemical composition of diets: EM, MJ/kg DM = 0.0205 × DP + 0.0398 × DF + 0.0173 × S + 0.0160 × C + 0.0147 × DOM where: DP—digestible protein, DF—digestible fat, S—starch, C—sugars, DOM—digestible organic matter (g/kg DM).; * Premix—mineral and vitamin premix content (per 1 kg diet): Choline chloride 400 mg, Fe 100 mg, Cu 160 mg, Co 0.4 mg, Mn 40 mg, Zn 140 mg, J 0.8 mg, Se 0.2 mg, antioxidant (Butylated hydroxyanisole, Butylated hydroxytoluene); IU: 12,000 Vit. A; 1500 Vit. D_3_; 70 mg Vit. E; 1.5 mg Vit. K_3_; 1.5 mg Vit. B_1_; 4 mg Vit. B_2_; 3 mg Vit. B_6_; 25 µg Vit. B_12_; 10 mg Pantothenic Acid; 20 mg Nicotinic acid; 2 mg folic acid; 100 µg Biotin; 0.9 g; 0.6 g; Calcium. Premix was produced by LNB.

**Table 4 animals-11-02894-t004:** The apparent ileal digestibility coefficients of protein and amino acids in raw and fermented lupine seeds and soybean meal (*n* = 5).

Item %	Component Effect							Fermentation Effect (F)			Seeds Effect (S)			Interaction (F × S)
	SBM	RNL	FNL	RYL	FYL	P	SEM	NO	YES	P	NL	YL	P	P
CP	74.88 ^a^	68.38 ^b^	73.47 ^a^	69.30 ^b^	70.20 ^b^	**0.011**	1.47	69.29	71.38	**0.041**	70.92	69.75	0.214	0.226
Lysine	77.80	71.29	75.33	71.64	71.44	0.116	0.36	71.48	73.60	0.371	73.54	71.55	0.408	0.325
Methionine	84.59 ^a^	63.61 ^b^	62.44 ^b^	64.72 ^b^	56.76 ^b^	**0.001**	2.26	63.78	59.91	0.169	62.51	61.18	0.535	0.187
Threonine	66.65 ^a^	52.48 ^b^	66.08 ^a^	48.47 ^b^	52.84 ^b^	**0.001**	1.84	50.25	60.20	**0.005**	60.03	50.41	**0.001**	**0.038**
Isoleucine	76.59 ^a^	73.61 ^ab^	75.38 ^a^	69.26 ^bc^	65.92 ^c^	**0.001**	1.05	71.19	71.18	0.633	74.60	67.77	**0.001**	0.133
Leucine	73.02 ^a^	66.66 ^b^	72.56 ^a^	69.81 ^ab^	66.10 ^b^	**0.010**	0.85	68.41	69.69	0.489	69.94	68.16	0.299	**0.007**
Valine	71.83 ^a^	63.17 ^ac^	68.19 ^ab^	58.81 ^cd^	56.06 ^d^	**0.001**	1.44	60.75	62.80	0.556	65.96	57.59	**0.001**	0.058
Histidine	69.87 ^a^	62.56 ^b^	69.22 ^a^	56.41 ^b^	61.79 ^b^	**0.001**	1.34	59.40	66.26	**0.007**	63.53	62.13	0.767	**0.014**
Arginine	83.57 ^bc^	85.26 ^b^	89.22 ^a^	80.46	83.68^bc^	**0.002**	0.85	82.60	86.76	**0.024**	87.46	81.89	**0.003**	0.798
Cystine	76.03 ^b^	82.10 ^a^	80.01 ^ab^	85.87 ^a^	82.63 ^a^	**0.016**	0.92	83.64	81.18	0.232	80.94	83.87	0.142	0.966
Phenylalanine	76.81	68.26	74.66	69.48	68.08	0.084	1.27	68.93	71.73	0.374	71.81	68.86	0.342	0.174
Asparagine	72.83 ^a^	65.23 ^b^	72.12 ^a^	59.90 ^b^	62.36 ^b^	**0.001**	1.37	62.27	67.78	**0.036**	69.06	60.99	**0.002**	0.289
Serine	73.62 ^a^	67.67 ^b^	75.54 ^a^	67.81 ^b^	66.61 ^b^	**0.002**	1.03	67.75	71.57	0.070	72.05	67.28	**0.022**	**0.019**
Tyrosine	71.27 ^a^	59.12 ^c^	68.48 ^ab^	55.35 ^c^	61.43 ^bc^	**0.002**	1.65	57.03	65.35	**0.012**	64.32	58.05	0.064	0.551
Proline	59.89	62.97	60.82	55.31	52.34	0.794	2.69	59.14	59.58	0.729	61.89	53.83	0.291	0.956
Glutamine	83.00	82.06	56.28	80.61	82.72	0.114	0.74	81.25	84.70	0.073	84.40	81.55	0.147	0.528
Glycine	59.05	53.45	57.24	49.07	51.83	0.487	1.92	51.01	54.83	0.496	55.55	50.29	0.314	0.914
Alanine	68.71 ^a^	58.72 ^bc^	60.81 ^b^	56.22 ^bc^	50.81 ^c^	**0.002**	1.60	57.33	56.36	0.548	59.88	53.81	**0.036**	0.186

^a–d^—data significantly different among groups at *p* < 0.05; *p*-values < 0.05 were signed in bold; SBM—diet contained soybean meal as the sole component high in protein; RYL—diet contained raw yellow lupine seeds; FYL—diet contained fermented yellow lupine seeds; RNL—diet contained raw narrow-leaved lupine seeds; FNL—diet contained fermented narrow-leaved lupine seeds.

**Table 5 animals-11-02894-t005:** The microflora content in the ileal digesta (*n* = 5).

log_10_ CFU/g	Component Effect							Fermentation Effect (F)			Seeds Effect (S)			Interaction (F × S)
	SBM	RNL	FNL	RYL	FYL	P	SEM	NO	YES	P	NL	YL	P	P
Total bacteria	8.12 ^b^	8.43 ^b^	8.47 ^b^	8.26 ^b^	10.99 ^a^	**0.001**	0.259	8.35	9.87	**0.001**	8.45	9.78	**0.001**	**0.001**
Yeast	4.60 ^bc^	5.04 ^abc^	6.14 ^a^	4.32 ^c^	5.72 ^ab^	**0.042**	0.224	4.68	5.91	**0.023**	5.59	5.10	0.262	0.749
Moulds	0.54 ^b^	0.85 ^b^	4.97 ^a^	3.56 ^a^	0.00 ^b^	**0.001**	0.446	2.20	2.21	0.994	2.91	1.58	**0.022**	**0.001**
LAB	8.20	8.12	8.52	8.55	7.47	0.204	0.163	8.33	7.93	0.319	8.32	7.95	0.359	**0.041**
Enterobacteriaceae	6.95	7.51	7.89	7.52	6.54	0.140	0.187	7.52	7.14	0.435	7.70	6.98	0.092	0.089

^a–c^—data significantly different among groups at *p* < 0.05; *p*-values < 0.05 were signed in bold; SBM—diet contained soybean meal as the sole component high in protein; RYL—diet contained raw yellow lupine seeds; FYL—diet contained fermented yellow lupine seeds; RNL—diet contained raw narrow-leaved lupine seeds; FNL—diet contained fermented narrow-leaved lupine seeds.

**Table 6 animals-11-02894-t006:** Dry matter, pH, ammonia content and volatile fatty acids content and profile in ileal digesta of pigs (*n* = 5).

Item	Component Effect							Fermentation Effect (F)			Seeds Effect (S)			Interactionb (F × S)
	SBM	RNL	FNL	RYL	FYL	P	SEM	NO	YES	P	NL	YL	P	P
Dry mater, %	7.95 ^c^	12.03 ^a^	10.84 ^b^	11.77 ^ab^	11.24 ^ab^	**0.001**	0.16	11.92	11.04	**0.012**	11.46	11.50	0.898	0.884
pH	6.54 ^b^	6.76 ^a^	6.72 ^a^	6.05 ^c^	6.52 ^b^	**0.001**	0.02	6.45	6.62	**0.010**	6.74	6.33	**0.001**	0.111
Ammonia, mmol/g	15.66 ^c^	23.85 ^a^	21.67 ^b^	24.89 ^a^	20.15 ^b^	**0.001**	0.30	24.37	20.91	**0.001**	22.76	22.52	0.738	0.652
VFA mmol/g	19.57 ^c^	40.11 ^b^	50.71 ^a^	50.85 ^a^	36.86 ^b^	**0.001**	1.45	45.48	43.78	0.682	44.41	43.86	0.707	0.254
Acetate, % VFA	81.38 ^a^	74.12 ^b^	75.81 ^ab^	75.47 ^ab^	71.50 ^b^	**0.029**	0.94	74.80	73.65	0.604	74.96	73.49	0.505	0.321
Propionate % VFA	13.89	19.66	17.63	15.74	18.97	0.107	0.74	17.70	18.30	0.725	18.64	17.35	0.454	0.188
Isobutyrate, % VFA	0.23 ^c^	0.21 ^c^	0.30 ^bc^	0.34 ^ab^	0.40 ^a^	**0.001**	0.01	0.27	0.35	**0.011**	0.25	0.37	**0.001**	0.401
Butyrate, % VFA	3.70 ^c^	4.27 ^bc^	4.66 ^bc^	6.35 ^ab^	7.13 ^a^	**0.018**	0.366	5.31	5.89	0.505	4.46	6.74	**0.011**	0.314
Isovalerate, % VFA	0.19 ^c^	0.36 ^b^	0.52 ^a^	0.34 ^b^	0.55 ^a^	**0.001**	0.02	0.31	0.54	**0.001**	0.44	0.45	0.898	0.228
Valerate, % VFA	0.62	1.38	1.08	1.76	1.45	0.114	0.14	1.57	1.26	0.357	1.23	1.60	0.257	0.455

^a–c^—data significantly different among groups at *p* < 0.05; *p*-values < 0.05 were signed in bold; SBM—diet contained soybean meal as the sole component high in protein; RYL—diet contained raw yellow lupine seeds; FYL—diet contained fermented yellow lupine seeds; RNL—diet contained raw narrow-leaved lupine seeds; FNL—diet contained fermented narrow-leaved lupine seeds.

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
