# Peer review of "The Effect of Yeast Fermentation of Two Lupine Species on the Digestibility of Protein and Amino Acids, Microflora Composition and Metabolites Production in the Ileum of Growing Pigs"

_animals, 2021, doi:10.3390/ani11102894_

Round 1

Reviewer 1 Report

Reviewer's remarks / comments

Comment 1 concerns the chapter Results, line 189-191

Table 4 contains a lot of figures and it is very difficult for the reader to perceive. I propose that in Table 4, under the index CP, the amino acids limiting the protein for pigs should be listed, and only then the remaining amino acids

Comment 2 concerns the chapter Results, line 229

I recommend that in the header of Table 6 the analyzed indicators should be presented in the same order as they are already given in the table itself in the "Item" column

The comment 3 concerns the chapter Summary, line 320

I recommend replacing Summary with Conclusions.

The comment 4 concerns the chapter Summary, line 324-326

The given statement requires clarification with the indication of amino acids (name the amino acids) for which digestibility increased.

The comment 5 concerns the chapter Summary

There is no practical conclusion (comment) based on the obtained results, namely what to do next with legume seeds fermentation (including lupins) and what is its significance for pig breeders.

Author Response

Dear Reviewer,

first of all, we would like to thank for comments and for considering a revised version of our manuscript. All changes made in the manuscript are marked in red. The manuscript in the revised form has been approved by all the co-authors.

Comment 1

concerns the chapter Results, line 189-191. Table 4 contains a lot of figures and it is very difficult for the reader to perceive. I propose that in Table 4, under the index CP, the amino acids limiting the protein for pigs should be listed, and only then the remaining amino acids.

Response: it was done.

Comment 2

concerns the chapter Results, line 229. I recommend that in the header of Table 6 the analyzed indicators should be presented in the same order as they are already given in the table itself in the "Item" column.

Response: it was done.

The comment 3

concerns the chapter Summary, line 320. I recommend replacing Summary with Conclusions.

Response: it was changed.

The comment 4

concerns the chapter Summary, line 324-326. The given statement requires clarification with the indication of amino acids (name the amino acids) for which digestibility increased.

Response: it was added.

The comment 5 concerns the chapter Summary

There is no practical conclusion (comment) based on the obtained results, namely what to do next with legume seeds fermentation (including lupins) and what is its significance for pig breeders.

Response: it was added: Using of Candida utilis yeast for fermentation of narrow-leafed lupine seeds could be a good solution for improving of feed quality and its nutritional value, but it could in-crease the cost of the nutrition, so more research including fermentation in other conditions as so as economic analysis of feeding should be recognized.

We hope that the changes made have improved the quality of the manuscript. We would be pleased if the revised manuscript could be published in MDPI Animals.

Yours sincerely

Corresponding author: Małgorzata Kasprowicz-Potocka

Reviewer 2 Report

The manuscript, # animals-1385378, The Digestibility of Protein and Amino Acids Fermented by Candida utilis in Yellow and Narrow-Leaved Lupine Seeds and Their Effect on the Microflora Composition and Metabolites Production in the Ileum of Growing Pigs, by Kasprowicz-Potocka1 et al. investigated the effects of raw and Candida utilis-fermented yellow and narrow-leaved lupine seeds on the apparent ileal digestibility coefficients of protein and amino acids in pigs, and the metabolic activity of their intestinal microflora. It can be considered for publication after the following major revisions.

Major comments

  1. This study was to investigate the effects of raw and Candida utilis-fermented yellow and narrow-leaved lupine seeds on the apparent ileal digestibility coefficients of protein and 17 amino acids with five cannulated male pigs. Why the author did not determine the SID (standard ileal digestibility) of protein and amino acids in different ingredients?
  2. The five replications in each treatment were insufficient. How did the authors keep effective number of replications for statistic analysis if outliers appeared?
  3. In statistic analysis of Table 4, 5, 6, the interaction (F × S) was showed. How did the authors explain these results which p-value of interaction (F × S) was less than 0.05?
  4. The authors explained that the content of alkaloids increased in fermented seeds because alkaloids were resistant to yeast fermentation. However, the content of alkaloids in fermented seeds was about three times higher than its in raw seeds. Did the author consider that some other reasons were responsible for the increase of alkaloids?
  5. In table 2, the counts of lactic acid bacteria were higher than total bacteria, which was not logical and hard to understand. Please double check the data and explain.
  6. Please double check whether the fermentation increased lysine content in both lupine species.

Minor comments

  1. The title of this manuscript was a little long. Please consider it.
  2. Line 390 and 406: The title of these references should be just capitalized the first word. Please consider them.
  3. The amino acids in this manuscript were showed in both full name and abbreviation. Please revised them following the author guidelines.

Author Response

Dear Reviewer,

first of all, we would like to thank for comments and for considering a revised version of our manuscript. All changes made in the manuscript are marked in red. The manuscript in the revised form has been approved by all the co-authors.

Comments and Suggestions for Authors

The manuscript, # animals-1385378, The Digestibility of Protein and Amino Acids Fermented by Candida utilis in Yellow and Narrow-Leaved Lupine Seeds and Their Effect on the Microflora Composition and Metabolites Production in the Ileum of Growing Pigs, by Kasprowicz-Potocka1 et al. investigated the effects of raw and Candida utilis-fermented yellow and narrow-leaved lupine seeds on the apparent ileal digestibility coefficients of protein and amino acids in pigs, and the metabolic activity of their intestinal microflora. It can be considered for publication after the following major revisions.

Major comments

  1. This study was to investigate the effects of raw and Candida utilis-fermented yellow and narrow-leaved lupine seeds on the apparent ileal digestibility coefficients of protein and 17 amino acids with five cannulated male pigs. Why the author did not determine the SID (standard ileal digestibility) of protein and amino acids in different ingredients?

Response: As we signed in the line 107-110 “ All diets were semi-synthetic and contained only one source of protein: soybean meal (SBM), raw yellow (RYL) or fermented yellow lupine seeds (FYL), raw narrow-leaved (RNL) or fermented narrow-leaved lupine seeds (FNL) at an amount resulting from the protein content in the components”. So we could only analyze these 5 protein sources because the other components had no protein and amino acids.

  1. The five replications in each treatment were insufficient. How did the authors keep effective number of replications for statistic analysis if outliers appeared?

Response: That’s true. We had 5 replications and this number is low, but those were canulated pigs and the procedure was very expensive. There were no outliers.

  1. In statistic analysis of Table 4, 5, 6, the interaction (F × S) was showed. How did the authors explain these results which p-value of interaction (F × S) was less than 0.05?

Response: The interaction was signed in: Table 4 for AID of threonine, leucine, histidine, and serine and 5 for: total bacteria, Lab and moulds but it is not known if these interactions are positive or negative. The used protein raw materials differed in microbiological purity both between cultivars and raw and fermented seeds, which could have influenced the interaction of both factors. It has been found that fermented seeds generally have a higher digestibility of protein and some amino acids and possibly other ingredients as they have been "shredded" during the fermentation process and may provide a better medium for bacteria as well as provide better access of digestive enzymes to the particles. These may affect both the digestibility of ingredients and the number of microorganisms.

  1. The authors explained that the content of alkaloids increased in fermented seeds because alkaloids were resistant to yeast fermentation. However, the content of alkaloids in fermented seeds was about three times higher than its in raw seeds. Did the author consider that some other reasons were responsible for the increase of alkaloids?

Response: We analyzed total content of alkaloids and the content of these substances was very low in the seeds, so firstly during fermentation some structural changes could exist (as complexation or creating of other forms of alkaloids). Secondly, during fermentation carbohydrates content, especially RFOs, is reduced by yeast what resulted in the higher content of other nutrients as protein and amino acids. It is clear that the higher content of alkaloids is connected with these changes but because of low content of alkaloids it is not proportional value, when you compare with other nutrients.

  1. In table 2, the counts of lactic acid bacteria were higher than total bacteria, which was not logical and hard to understand. Please double check the data and explain.

Response: It was a mistake. The Lactic acid bacteria in the fermented narrow-leaved lupine seeds should be 4.8∙105±3.1∙103 It was improved.

  1. Please double check whether the fermentation increased lysine content in both lupine species.

Response: It was checked. The content of Lysine in narrow-leaved lupine seeds increased from 1.29% to 1.54% and for yellow lupine seeds from 1.65% to 2.02%

  1. The title of this manuscript was a little long. Please consider it.

Response: Title was changed: The Effect of Yeast Fermentation of Two Lupine Species on the Digestibility of Protein and Amino Acids, Microflora Composition and Metabolites Production in the Ileum of Growing Pigs.

  1. Line 390 and 406: The title of these references should be just capitalized the first word. Please consider them.

Response:  All the titles start from capital letter.

  1. The amino acids in this manuscript were showed in both full name and abbreviation. Please revised them following the author guidelines.

Response: it was changed.

We hope that the changes made have improved the quality of the manuscript. We would be pleased if the revised manuscript could be published in MDPI Animals.

Yours sincerely

Małgorzata Kasprowicz-Potocka

Round 2

Reviewer 2 Report

Regarding microbial analyses, Please add the method for analyzing mould content.

Author Response

Dear Reviewer,

All changes made in the manuscript are marked in red.

The yeast spores and molds were identified together on the basis of morphologic features in accordance with the methodology of determinations. I was added in line 172.

We hope that the changes made have improved the quality of the manuscript. 

Yours sincerely

Małgorzata Kasprowicz-Potocka